# Biogeochemical Shifts During Arctic Spring: Potential Reduction of CH$_4$ and N$_2$O Emissions Driven by Surfactants in the Sea-Surface Microlayer

Lina A. Holthusen[1,2], Hermann W. Bange[2], Thomas H. Badewien[1], Julia C. Muchowski[6], Tina Santl-Temkiv[7,8], Jennie Spicker Schmidt[7,8], Oliver Wurl[1], Damian L. Arévalo-Martínez[2,3,4,5]

[1] Institute for Chemistry and Biology of the Marine Environment, Carl von Ossietzky Universität Oldenburg, Wilhelmshaven, Germany.

[2] Chemical Oceanography Research Unit, GEOMAR Helmholtz Centre for Ocean Research Kiel, Kiel,
Germany.

[3] Institute of Geosciences, Kiel University, Kiel, Germany.

[4] Department of Microbiology, Radboud University, Nijmegen, Netherlands.

[5] Marine Chemistry Department, Leibniz Institute for Baltic Sea Research Warnemünde, Rostock, Germany.

[6] Swedish Polar Research Secretariat, Luleå, Sweden.

[7] Department of Biology, Microbiology, Aarhus University, Aarhus, Denmark.

[8] Arctic Research Centre, Aarhus University, Aarhus, Denmark.

*Correspondence to*: Lina A. Holthusen (lina.aleke.holthusen@uol.de)

**Abstract.** The sea-surface microlayer (SML) is the less than one millimeter thin interface between the surface ocean and the overlying atmosphere and plays a crucial role in sea-air gas exchange processes. However, its role in sea-air exchange processes in sea ice-influenced environments such as the Arctic Ocean remains poorly understood. Here, we present the first *in situ* measurements of surfactants accumulating in the SML of the Fram Strait, coupled with near-surface measurements of the climate-relevant trace gases $CH_4$ and $N_2O$. Sampling of the undisturbed surface ocean was conducted at leads and ice holes during the onset of sea ice melt and an early algal bloom between May and June 2023. Our results reveal that the region acted as a minor source of $CH_4$ and $N_2O$. Nonetheless, the algal bloom stimulated the production of surfactants, which accumulated in the SML of open leads, potentially reducing the emissions of $CH_4$ and $N_2O$ to the atmosphere. These findings highlight the importance of resolving short-term surface processes during seasonal transitions and of integrating SML dynamics into investigating trace gas fluxes in polar regions.

## 1. Introduction

Climate-relevant gases, such as methane ($CH_4$) and nitrous oxide ($N_2O$), are the most abundant greenhouse gases in the atmosphere after $CO_2$ and play an important role in the global climate due to their high global warming potential (IPCC, 2021). While the ocean acts as a minor source for atmospheric $CH_4$, it is a major source for $N_2O$ (Saunois et al., 2025; Tian et al., 2024). The emission estimates of both gases are subject to significant uncertainties, both due to large temporal and spatial variability, and limited data coverage (Weber et al., 2019; Yang et al., 2020).

Oceanic $CH_4$ emissions account for 1–3% of the global $CH_4$ budget (Saunois et al., 2025). The surface ocean is typically supersaturated, whereas the deep ocean is in equilibrium or undersaturated with $CH_4$ relative to the atmosphere (Reeburgh, 2007). $CH_4$ can be released by diffusion from sediments, gas seeps, melting gas hydrates, or thawing subsea permafrost (Barnes and Goldberg, 1976; James et al., 2016). $CH_4$ concentrations can rapidly decrease due to microbial anaerobic and aerobic oxidation of $CH_4$ in sediments and the entire water column, respectively (Venetz et al., 2024). Due to this so-called microbial methane filter, a large quantity of $CH_4$ originating from the sea floor does not reach the sea surface. Aerobic *in situ* production in the well-oxygenated surface layer of the ocean can lead to a supersaturation with $CH_4$. This production has been reported in several studies, which suggest *in situ* production by bacteria during the degradation of organic material (e.g., Repeta et al., 2016), particularly during phosphate or nitrate limitation (Damm et al., 2010; Karl et al., 2008). Another source can be direct production by phytoplankton (Mao et al., 2024), cyanobacteria (Bižić et al., 2020), and during zooplankton grazing (Stawiarski et al., 2019). A potential abiotic production pathway for $CH_4$ is the photodegradation of chromophoric dissolved organic matter (CDOM), which occurs in the upper meters of the water column (Li et al., 2020).

The world's oceans contribute approximately 26–31% of the global $N_2O$ emissions, with high temporal and spatial variability (Yang et al., 2020; Tian et al., 2024). Oceanic $N_2O$ is mainly produced by microbial nitrification and denitrification (Bakker et al., 2014). Nitrification dominates under oxic conditions, while denitrification dominates under suboxic or anoxic conditions and accounts for approximately 7–20% of the total production of marine $N_2O$ (Bakker et al., 2014; Ji et al., 2018). During nitrification, $N_2O$ is a by-product of ammonia ($NH_3$) oxidation to nitrite ($NO_2^-$), which is mainly carried out by ammonia-oxidizing archaea (Löscher et al., 2012) and ammonia-oxidizing bacteria (Bakker et al., 2014). Denitrification is the step-wise microbial reduction of nitrate ($NO_3^-$) to dinitrogen ($N_2$) in $O_2$-depleted seawater (Bakker et al., 2014). Here, $N_2O$ is an intermediate product, and further reduction of $N_2O$ to $N_2$ is effectively inhibited by the presence of $O_2$ (Dalsgaard et al., 2014). An abiotic source of

N$_2$O in the marine environment is the photochemical degradation of nitrite, which can yield higher N$_2$O production rates than biological processes (Leon-Palmero et al., 2025).

In the Arctic Ocean (AO), measurements of dissolved CH$_4$ and N$_2$O concentrations are limited, yet existing observations reveal pronounced spatial and temporal variability (e.g., Rees et al., 2022). Direct measurements of CH$_4$ sea-air fluxes demonstrate that the Central AO is a small source of atmospheric CH$_4$ during summer (average 3.5 µmol m$^{-2}$ d$^{-1}$, approximately 0.009 Tg yr$^{-1}$) (Prytherch et al., 2024) compared to global oceanic emissions of 6-12 Tg yr$^{-1}$ (Weber et al., 2019). In contrast, shelf regions such as the Laptev Sea (0.83 Tg yr$^{-1}$), East Siberian Sea (0.62 Tg yr$^{-1}$), and Chukchi Sea (0.03 Tg yr$^{-1}$) show significantly higher and more variable CH$_4$ fluxes, with local peaks of up to 38.5 µmol m$^{-2}$ d$^{-1}$ (Thornton et al., 2020). Suggested sources include the release of CH$_4$-enriched brine from sea ice (Damm et al., 2015a), *in situ* production via methylotrophic methanogenesis with DMSP (dimethylsulfoniopropionate) as a precursor (Damm et al., 2015b), and release from the sea floor (e.g., Cramm et al., 2021). Most studies investigating dissolved N$_2$O in the AO found surface waters near atmospheric equilibrium, with a trend of higher emissions in shelf areas compared to the open ocean (Verdugo et al., 2016; Zhan et al., 2021; Muller et al., 2024). N$_2$O dynamics in the AO are driven by nitrification and denitrification in sediments and the water column, respectively (Fenwick et al., 2017; Kitidis et al., 2010). Sea ice was observed to be undersaturated in N$_2$O (Randall et al., 2012), and its meltwater may dilute surface concentrations (Kitidis et al., 2010; Fenwick et al., 2017).

The sea-surface microlayer (SML) describes the less than 1 mm thick boundary layer between the ocean and the atmosphere (Liss & Duce, 1997) and is characterized by the accumulation of surface-active substances (SAS), so-called surfactants (Wurl et al., 2009; Cunliffe et al., 2013). Organic material, mainly derived from phytoplankton and bacteria, can accumulate in the SML and create biofilm-like habitats, known as slicks (Wurl et al., 2016; Ẑutić et al., 1981). In polar environments, extracellular polymeric substances (EPS) are an important component of the SAS accumulating in the SML, where they contribute to the formation of biofilms (Gao et al., 2012; Orsi et al., 1995). EPS are produced by phytoplankton, primarily diatoms, and by bacteria as cryoprotectants and are therefore abundant in sea ice and brine (Aslam et al., 2012; Krembs et al., 2002; Underwood et al., 2013). During melting, EPS are released into the surface ocean, providing a source of organic carbon (Riebesell et al., 1991; Riedel et al., 2006). Due to cross-linking of their polymers, which mainly consist of polysaccharides, EPS can form marine gels and aggregates that influence particle sinking rates and act as potential sources of cloud condensation nuclei (CCN) (Orellana et al., 2011; Riebesell et al., 1991; Verdugo, 2012). Additionally, these aggregates can serve as hotspots of microbial activity (Simon et al., 2002). Because SAS-enrichment in the SML builds a physico-chemical barrier and influences hydrodynamics, small-scale turbulences, and wind-wave interactions, it plays a crucial role in sea-air interactions of gases (McKenna and McGillis, 2004; Pereira et al., 2016; Mustaffa et al., 2020). Studies investigating SAS concentrations in the SML of the AO are scarce, but the presence of an SML in lead waters was observed by Knulst et al. (2003).

Data coverage for dissolved CH$_4$ and N$_2$O in the AO, including the Fram Strait, is sparse, particularly for the spring-summer transition and the onset of sea ice melt. Additionally, the characteristics of the SML in the AO are highly understudied, despite its high potential for shaping emissions of climate-relevant trace gases. The onset of sea ice melt is a key period, as CH$_4$ and N$_2$O could accumulate in brine channels within sea ice and subsequently escape to the ocean and atmosphere as the ice melts (Damm et al., 2015a; Randall et al., 2012). Additionally, sea ice melt triggers significant shifts in hydrographic and biogeochemical conditions, including the development of

under-ice algal blooms and surface slicks enriched in sea ice-derived organic matter (Assmy et al., 2017; Willis et al., 2023). These changes have the potential to alter trace gas dynamics substantially. With Arctic warming progressing rapidly and accelerating sea ice loss (Huang et al., 2017; Smedsrud et al., 2022), gaining insight into these processes is essential for improving predictions of greenhouse gas emissions and their feedbacks in the Arctic climate system. In this study, we address this significant knowledge gap by presenting the first observations of dissolved $CH_4$ and $N_2O$ dynamics in open leads and under-ice water, linked to SAS-enrichment in the SML during the onset of sea ice melt in the Fram Strait in spring and early summer 2023.

## 2. Methods

### 2.1 Study Area

The fieldwork was conducted in May and June 2023 on board the Swedish icebreaker (IB) ODEN during the Expedition ARTofMELT23. The study area in the northern Fram Strait, located between Greenland and Svalbard, is exposed to extreme atmospheric warming, with average winter temperatures on Svalbard in recent years more than 7°C warmer compared to the 1970s (Bilt et al., 2019). The Fram Strait is an important gateway for water mass exchange between the AO and the North Atlantic Ocean. Warm and saline Atlantic Water enters the AO via the northward-flowing West Spitsbergen Current (WSC) on the eastern side of the Fram Strait, while colder and fresher Polar Water flows southwards with the East Greenland Current (EGC) on the western side of the Fram Strait (Rudels et al., 2005; Beszczynska-Möller et al., 2012). During the last decades, the warming of the inflowing Atlantic Water into the AO has led to an elevated heat content and sea ice loss and consequently to an increased poleward expansion of the Atlantic Water in the Arctic Seas (Beszczynska-Möller et al., 2012; Smedsrud et al., 2022). Sampling and measurements were conducted at (i) six ice stations on drifting sea ice, (ii) two off-ship stations (accessed by helicopter) near the ice station to access open leads, and (iii) five CTD stations (Fig. 1 & Table 1). The sampling sites were located between the sea ice edge and approximately 130 km away from the ice edge. The first and last sampling stations of this study (10 May and 11 June) were located close to the ice edge (see supplementary, Fig. S1).

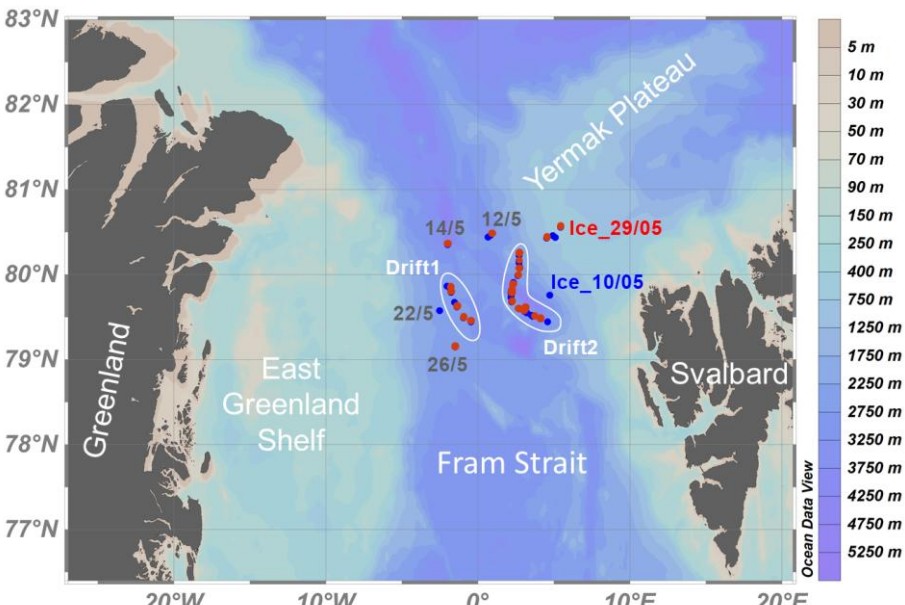

*Figure 1: Study area in the Fram Strait between Greenland and Svalbard. White bordered regions indicate IB ODEN's position during the two ice drift stations. Red dots indicate VMP stations, and blue dots indicate CTD stations. Grey labels indicate stations without $CH_4$ and $N_2O$ data but CTD and/or VMP measurements.*

*Table 1: Overview of sampling stations and locations, daily mean meteorological conditions (Murto et al., 2024), and chlorophyll concentrations from VMP data (average of the upper 5 m).* = sampled on the Yermak Plateau, ° = sampled at off-ship stations accessed by helicopter.*

| Date | 10 May | 18 May | 20 May | 29 May | 1 June | 4 June | 5 June | 6 June | 8 June | 9 June | 11 June |
|---|---|---|---|---|---|---|---|---|---|---|---|
| Station | Ice _10/05 | Drift1 _18/05 | Drift1 _20/05 | Ice _29/05 | Drift2 _01/06 | Drift2 _04/06 | Drift2 _05/06 | Drift2 _06/06 | Drift2 _08/06 | Drift2 _09/06 | Drift2 _11/06 |
| Sampling | CTD | CTD | Lead | Lead | Under Ice + CTD | Under Ice | Lead° + CTD | Under Ice | Lead° | CTD | Under Ice |
| Water Mass | WSC | EGC | EGC | WSC* | WSC | WSC | WSC | WSC | WSC | WSC | WSC |
| Air Temp [°C] | -1.56 | -5.34 | -0.94 | -4.14 | -5.28 | -3.6 | -4.21 | -2.91 | -3.07 | -3.52 | 1.2 |
| Wind speed [m s$^{-1}$] | 4.9 | 5.9 | 6.1 | 7.9 | 5 | 4.8 | 4.4 | 3.5 | 3 | 4.2 | 5.4 |
| Solar Radiation [W m$^{-2}$] | 102 | 262 | 134 | 281 | 349 | 260 | 242 | 335 | 198 | 239 | 138 |
| Chlorophyll [μg L$^{-1}$] | < 1 | < 1 | < 1 | 2.1 | 8.3 | 8.1 | 10.3 | 8.1 | 6.8 | 11.8 | 8.1 |

## 2.2 Sampling

During both ice and off-ship stations, discrete water samples were collected at a reasonable distance from the ship to minimize its influence on sampling. Water was retrieved either from leads (over the ice edge), or from ice holes, which were manually sawed and maintained during the entire drift stations. Sampling at the leads was always

carried out downwind of the lead. Discrete surface water samples for $CH_4$ and $N_2O$ were collected using a Van-Dorn water sampler (11.100-Van Dorn 2 L, KC Denmark), which was closed right below the surface, capturing the upper 30 cm of the water column. Samples for the entire water column (3–2800 m) were retrieved from Niskin bottles during CTD casts.

## 2.3 CH$_4$ and N$_2$O

Trace gas samples were collected during five CTD casts and eight ice stations (four at leads and four from under-ice water). Samples were obtained bubble-free and in triplicate in 50 mL vials for $CH_4$ and 20 mL vials for $N_2O$ and sealed airtight. Samples from the ice and off-ship stations were kept cold and in the dark and returned to the ship within a maximum of two hours after sampling. Back on board, they were immediately poisoned with 125 μL ($CH_4$) and 50 μL ($N_2O$) saturated aqueous solution of mercury chloride ($HgCl_2$). Samples collected during CTD

casts were poisoned immediately after sampling. The stations at which samples were taken both from CTD and ice stations (1 June and 5 June) show good agreement in gas concentration, despite the difference in the time elapsed between sampling and preservation (see Figure 4). After preservation, all samples were stored at room temperature in the dark until analysis at the University of Southern Denmark (Department of Biology, Nordcee) within 12 months after sampling. Both gas concentrations were measured using a static headspace method and gas

chromatography (Wilson et al., 2018). The procedure for $CH_4$ followed the method described in (Holthusen et al., 2025). For $N_2O$, a Thermo Scientific TRACE 1300 GC with an Electron Capture Detector and a 10 ppm $N_2O$ standard gas (Air Liquide, nitrous oxide N25, Helium N50) for calibration in different volumes (500, 250, 100, 50, 25, and 10 μL) was used. The mean coefficient of variation between triplicate measurements was +/- 5.5% and +/- 2.1% for $CH_4$ and $N_2O$, respectively. Please note that the trace gas concentrations from the ice and off-ship

stations might be affected by the time delay between sampling and poisoning caused by the transport to the ship. However, due to the cold temperatures during transport, microbial activities most probably were very low, and therefore, we assume no significant change in the dissolved trace concentrations during the transport.

For the subsequent calculation of concentrations and saturations, the method as described in Holthusen et al. (2025) was used. For $N_2O$, the solubility equation by Weiss & Price (1980) was used. For the calculation of sea-air fluxes ($F$), the parametrization of the gas transfer velocities normalized to $k_{660}$ for open water ($k_{open}$) and for fractional sea ice coverage ($k_{SIC}$) by Butterworth & Miller (2016) was used:

$$k_{open} = 0.245\,U_{10}{}^2 + 1.3$$

$$k_{SIC} = (1 - SIC)\,k_{open}$$

The daily mean sea ice coverage (SIC) was retrieved from E.U.-Copernicus Marine Service (2020, https://doi.org/10.48670/moi-00007) and the hourly mean wind speed corrected to 10 m ($U_{10}$) from Prytherch et al. (2024). For each sampling date, the closest available mean daily atmospheric mole fractions of $CH_4$ and $N_2O$ were obtained from the nearest monitoring site: Zeppelin Station in Ny-Alesund, Svalbard (University of Stockholm Meteorological Institute; NOAA ESRL GML CCGG Group, 2019). The resulting concentrations and sea-air fluxes are summarized in Table 2 (see Section 3.2).

**2.4 Hydrographic measurements**

During the ice and off-ship stations, hydrographic measurements were performed concurrently with discrete sampling using a Vertical Microstructure Profiler (VMP) (VMP250-IR, Rockland Scientific) equipped with a JAC CT temperature and conductivity sensor (JFE Advantech Co.), an FP07 fast temperature probe (Thermometrics), and a CLTU chlorophyll and turbidity sensor (JFE Advantech Co.). The VMP was manually deployed in ice holes or from leads in free fall down to 500 m and retrieved with an electrical winch. During CTD casts, a rosette equipped with 24 Niskin bottles (12 L each) and a SeaBird 911 CTD probe with dual SeaBird temperature (SBE 3), dual conductivity (SBE 04C), turbidity (WET Labs ECO), and fluorescence (WET Labs ECO-AFL/FL) sensors was deployed. Fluorescence data were used as an indicator of chlorophyll concentrations.

**2.5 Surfactants**

SAS samples were collected from both the SML and the underlying water (ULW) at 1 m depth. To collect water from the SML, the glass plate technique (Harvey and Burzell, 1972; Cunliffe and Wurl, 2014) was applied. A 40x30 cm glass plate was vertically introduced into the water and slowly removed (approximately 5 cm s$^{-1}$) (Cunliffe and Wurl, 2014). The SML water adhering to the glass plate was scraped off with a wiper into a sample vial, and the process was repeated until a sample volume of approximately 500 mL was collected. From this, one subsample was filtered through a 0.2 µm nucleopore syringe filter into a 50 mL plastic vial, and one subsample was filled directly into a second vial. Both sample types were stored in the dark at 4°C and analyzed within 4–6 weeks after sampling using alternating current voltammetry (VA Stand 747, Metrohm, Switzerland) with a hanging mercury drop electrode (Ćosović and Vojvodić, 1998; Rickard et al., 2019). In this method, SAS adsorb onto the surface of the mercury drop, leading to a reduction in the capacitive current. The total surface activity is determined from the magnitude of this decrease in current (Ćosović and Vojvodić, 1998). For comparability, the response is calibrated against a reference surfactant with known surface activity (Triton X-100, Sigma-Aldrich, Germany) as

an internal standard, and the results are expressed as SAS concentrations in Triton X-100 equivalents (µg Teq L$^{-1}$) (Sander and Henze, 1997).

Triplicate measurements of 13 samples with very low concentrations (<10 µg Teq L$^{-1}$) showed a high variation and were therefore excluded. The variation between the triplicate measurements of the remaining samples was on average +/- 10%. Please note that the sea ice edge around the ice holes (with an approximate size of 1 m$^2$) was most likely affecting the concentrations of SAS accumulating in the SML and thus introducing an unknown uncertainty in the measurements. A sample preservation experiment revealed that the time between sampling and measurement caused a decrease in SAS concentrations of approximately 5–21% (Fig. S2). In previous studies conducted under temperate conditions, SAS concentrations exceeding 200 µg Teq L$^{-1}$ have been associated with a reduction in gas transfer velocities by approximately 60%, while concentrations above 1000 µg Teq L$^{-1}$ have been classified as slicks (Mustaffa et al., 2020; Wurl et al., 2011). Maximum suppression of gas exchange rates was in a similar range in laboratory and wind-wave tunnel experiments using artificial monolayers and natural seawater (Brockmann et al., 1982; Broecker et al., 1978; Pereira et al., 2018; Ribas-Ribas et al., 2018; Salter et al., 2011). We used these values as a comparative threshold, as no Arctic-specific measurements are currently available. However, deviations from this relationship are possible due to potential differences in SAS composition and wind-wave interactions in Arctic leads. Even if SAS concentrations had been reduced by 21% because of the time lag between sampling and measurement, the sample classification would not have changed.

**2.6 Community composition analysis, nutrients, and meteorological data**

DNA was extracted using the DNeasy PowerSoil Pro Kit (Qiagen) from SML and ULW samples and concentrated on polycarbonate filters. Quantification of eukaryotic (18S rRNA) and bacterial (16S rRNA) gene copies by qPCR followed Wieber et al. (2025). Details of the laboratory procedure and additional data on microbial abundances can be found in the supplementary (Fig. S6). Concentrations of inorganic nutrients (NO$_x$ = sum of nitrate and nitrite; PO$_4^{3-}$ and NH$_4^+$) were taken from Rush & Vlahos (2025). Air temperature and wind speed were continuously measured at the foremast of IB ODEN (Prytherch et al., 2024) and solar radiation above the bridge (Murto et al., 2024).

**3. Results**

**3.1 Hydrography and Meteorology**

The collected hydrographic data showed clear signs of the EGC and the WSC (Fig. 2). The first sampling and measurements of this study (10 May) were conducted closest to Svalbard but off the shelf in the deep waters of the Fram Strait (Fig. 1 & Table 1). This water mass can be associated with the WSC. On 12 and 14 May, CTD and VMP measurements were conducted in the northern central part of the Fram Strait, marking the transition between the WSC and the EGC. Between 17 and 21 May (first drift station), the ship drifted southeast with the ice. Here, CTD/VMP casts and one ice station sampling were conducted in the waters of the EGC. From 24 to 27 May, a storm event occurred with wind speeds of up to 17 m s$^{-1}$. After a transit to the northeast of the Fram Strait, measurements and sampling were continued on 28/29 May on the Yermak Plateau north of Svalbard, in the waters of the WSC. Subsequently, the second drift station was established, and sampling was conducted regularly between 31 May and 11 June. A full description of the observed water masses and section plots for salinity and temperature for all VMP profiles are in the supplementary (Fig. S3). Recorded air temperatures (Fig. 3) showed a cold phase

between 11 and 18 May (<-10°C air temperature), followed by a warmer phase (-8 to 0.4°C) until 10 June. Afterwards, the air temperatures rose above 0°C, reaching 1.8°C on 12 June. However, melting of sea ice seemed to have started already on 5 June, as slush and meltwater appeared on top of the ice (observed during ice work on that day).

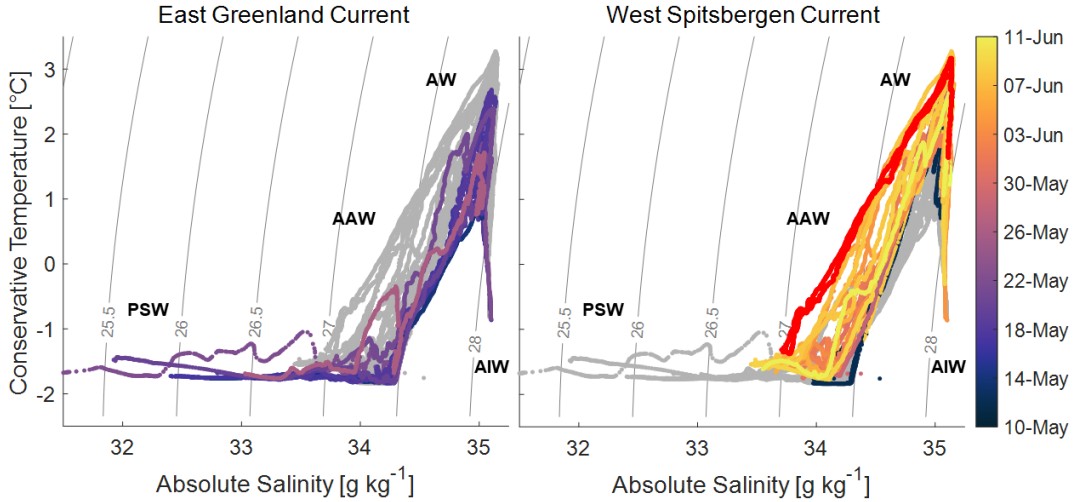

*Figure 2: Water mass distribution in the Fram Strait during the study period, identified by absolute salinity, conservative temperature, and density for the EGC (left) and WSC (right), based on all VMP and CTD measurements. Red data points indicate measurements from 5 June. PSW=Polar Surface Water, AW=Atlantic Water, AAW=Arctic Atlantic Water, AIW=Arctic Intermediate Water.*

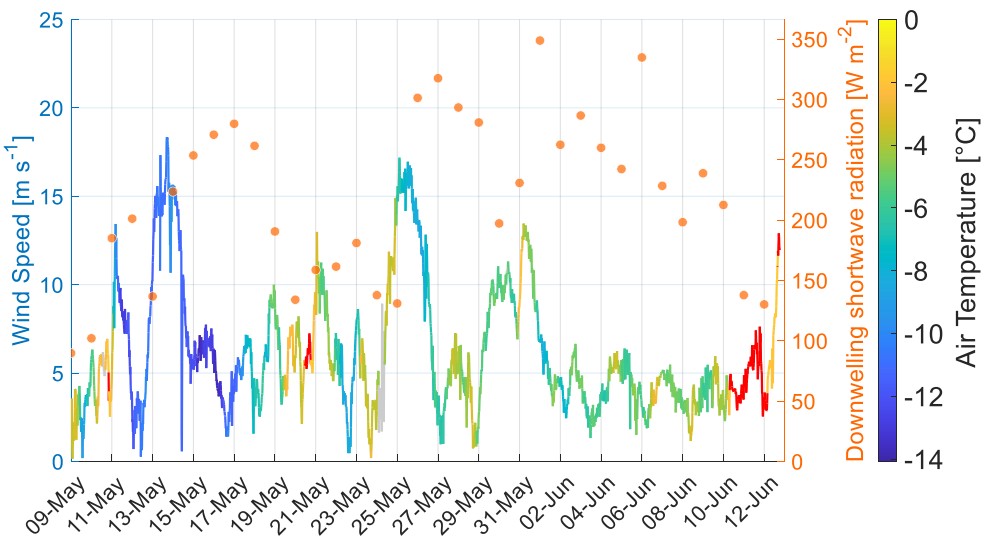

*Figure 3: Wind speed and air temperature (20 min average), and downwelling shortwave radiation (daily average) during the study period. Air temperatures over 0°C are marked in red. Data are from Prytherch et al. (2024) and Murto et al. (2024).*

**3.2 CH$_4$ and N$_2$O**

Dissolved surface CH$_4$ concentrations ranged from 3.7 to 6.0 nmol L$^{-1}$ with an average of 4.6±0.6 nmol L$^{-1}$ (mean ± standard deviation), corresponding to an average saturation of 110±14% (Fig. 4a). The highest concentration was measured on 5 June during an off-ship station in a lead, and the lowest on 11 June during an ice station in an ice hole. At the majority of the stations, seawater was oversaturated or close to equilibrium concentration, and thus

resulting in emissions of CH$_4$ from the ocean to the atmosphere. On 18 May, 8 June, and 11 June, negative fluxes and therefore CH$_4$ uptake by the ocean were observed. The average $F_{SIC}$ was 0.13±0.38 µmol m$^{-2}$ d$^{-1}$. Dissolved

surface $N_2O$ concentrations were above the atmospheric equilibrium concentration throughout this study and ranged from 18.2 to 21.1 nmol $L^{-1}$ with an average of 19.3±0.9 nmol $L^{-1}$, corresponding to an average saturation of 112±5% (Fig. 4b). The highest concentration was found in lead waters in the area of the Yermak Plateau (see Fig. 1), and the lowest during the second ice drift in both under-ice and lead water. Overall, $N_2O$ was emitted from the ocean to the atmosphere at all stations with an average $F_{SIC}$ of 0.96±0.99 µmol $m^{-2}$ $d^{-1}$.

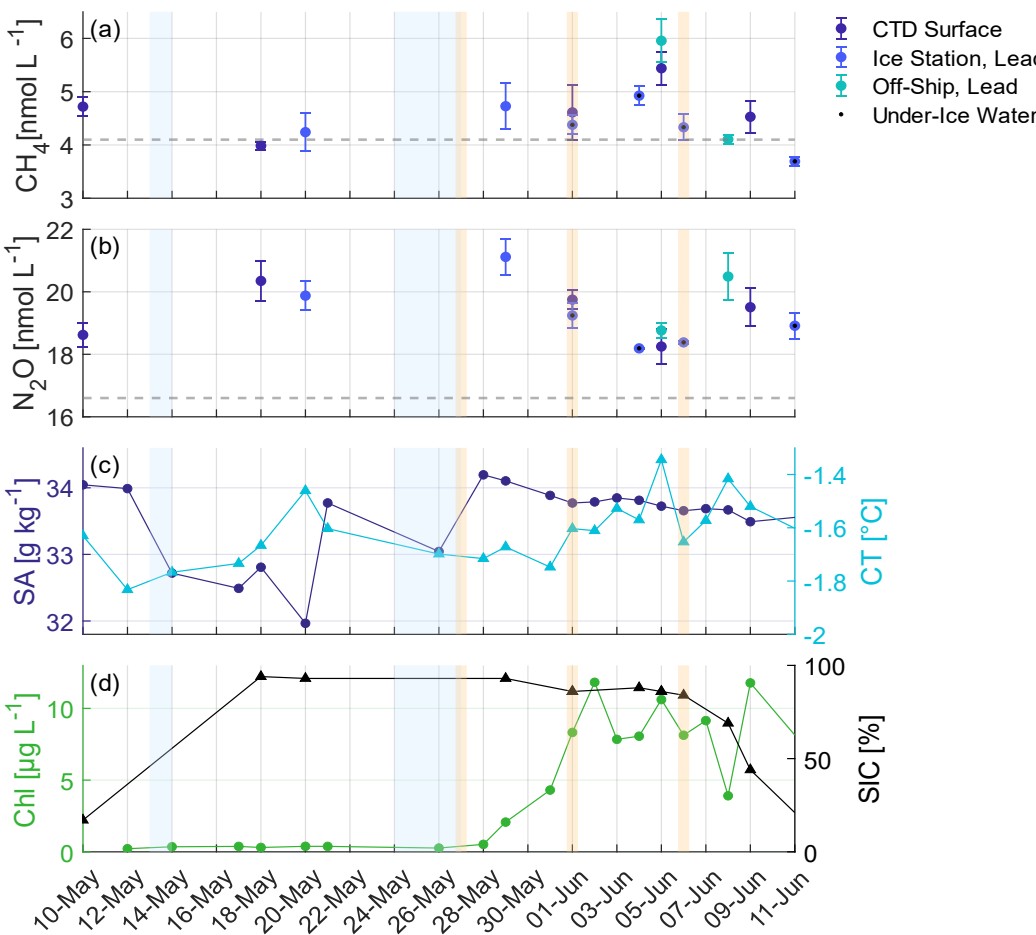

*Figure 4: Dissolved a) CH₄ and b) N₂O concentrations of surface samples from CTD, ice and off-ship stations. Dark blue dots indicate CTD surface samples (from 3 m water depth), bright blue dots indicate ice station samples from leads, dots with additional black dot from under-ice water (0.3 m water depth) and cyan dots indicate off-ship lead samples (0.3 m water depth). Error bars show the standard deviation of triplicate measurements of each sample. Grey dashed lines indicate the average equilibrium concentrations of the respective gas during the study. c) Mean value of the upper 5 m for Absolute Salinity (SA) and Conservative Temperature (CT) from VMP casts from 12 May to 11 June and from a CTD cast on 10 May. d) Chlorophyll (Chl, mean of the upper 5 m from VMP data) and Sea Ice Coverage (SIC) from E.U.-Copernicus Marine Service (2020, https://data.marine.copernicus.eu/-/9a661qmlrm). Blue shaded areas indicate storm events with wind speed exceeding 15 m s⁻¹, while orange shaded areas highlight days with mean daily solar radiation above 300 W m⁻².*

*Table 2: Overview of dissolved CH₄ and N₂O concentrations and comparison of sea-air fluxes considering open water ($F_{open}$) and sea ice coverage ($F_{SIC}$, SIC data from E.U.-Copernicus Marine Service, 2020). * = sampled on the Yermak Plateau, **sampled close to the ice edge, ° = off-ship stations.*

| Day | Position | $U_{10}$ [m s⁻¹] | SIC [%] | Diss. Conc. [nmol L⁻¹] | | $F_{open}$ [µmol m⁻² d⁻¹] | | $F_{SIC}$ [µmol m⁻² d⁻¹] | |
|---|---|---|---|---|---|---|---|---|---|
| | | | | CH₄ | N₂O | CH₄ | N₂O | CH₄ | N₂O |
| **10 May** | CTD** | 6 | 17 | 4.7 | 18.6 | 1.28 | 3.24 | 1.06 | 2.69 |
| **18 May** | CTD | 6.4 | 94 | 4.0 | 20.4 | -0.66 | 7.83 | -0.04 | 0.47 |
| **20 May** | Lead | 5.5 | 93 | 4.3 | 19.9 | 0.05 | 5.25 | 0.00 | 0.37 |
| **29 May** | Lead* | 9.3 | 93 | 4.7 | 21.1 | 2.94 | 20.7 | 0.21 | 1.45 |
| **1 June** | Under ice | 3.7 | 86 | 4.4 | 19.2 | 0.22 | 2.18 | 0.03 | 0.31 |
| **1 June** | CTD | 5.2 | 86 | 4.6 | 19.8 | 0.80 | 4.64 | 0.11 | 0.65 |
| **4 June** | Under ice | 5.6 | 88 | 4.9 | 18.2 | 1.63 | 2.03 | 0.20 | 0.24 |
| **5 June** | Lead° | 2.6 | 86 | 6.0 | 18.8 | 1.29 | 1.19 | 0.18 | 0.17 |
| **5 June** | CTD | 4 | 86 | 5.4 | 18.3 | 1.59 | 1.36 | 0.22 | 0.19 |
| **6 June** | Under Ice | 3.7 | 84 | 4.3 | 18.4 | 0.17 | 1.20 | 0.03 | 0.19 |
| **8 June** | Lead° | 1.3 | 69 | 4.1 | 20.5 | -0.02 | 1.41 | -0.01 | 0.44 |
| **9 June** | CTD | 5.6 | 44 | 4.5 | 19.5 | 0.78 | 4.90 | 0.44 | 2.74 |
| **11 June** | Under ice** | 5.4 | 21 | 3.7 | 18.9 | -0.96 | 3.22 | -0.76 | 2.54 |

### 3.3 Surfactants

SAS concentrations during this study showed a high variability depending on the sampling site (Fig. 5). Water samples from ice holes (4 June, 6, and 11) showed generally low concentrations with 19–89 µg Teq L$^{-1}$ in the SML, 13–80 µg Teq L$^{-1}$ in the ULW, and 4 out of 12 samples were below the detection limit (<10 µg Teq L$^{-1}$). Lead samples showed higher concentrations and ranged from 39–11788 µg Teq L$^{-1}$ in the SML and from 29–355 µg Teq L$^{-1}$ in the ULW (6 out of 16 samples were below the detection limit). On 5 June, the highest concentrations (11788 µg Teq L$^{-1}$) occurred with strong enrichment in the SML compared to the ULW (below detection limit). The filtered SML sample from this station exhibited a significantly lower concentration of 39 µg Teq L$^{-1}$, representing the highest difference between filtered and unfiltered samples observed in this study. At this station, floating slush ice from the lead surface was collected and showed a SAS concentration of 192 µg Teq L$^{-1}$. The second highest concentration was found in the SML on 29 May (1435 µg Teq L$^{-1}$) in a lead during a wind speed of 10 m s$^{-1}$. Only on 6 June (ice hole) and 8 June (lead), SAS concentrations in the ULW exceeded those in the SML.

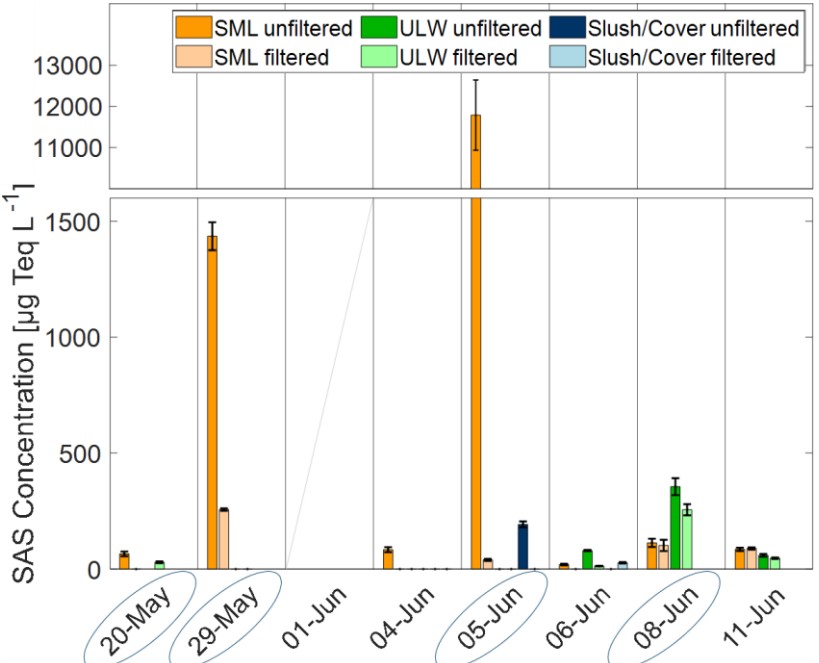

*Figure 5: SAS concentrations of filtered and unfiltered samples of SML, ULW, and slush ice at the surface. There is no SAS data available for 1 June. Blue ellipsoids mark open lead samples.*

### 4. Discussion

The water mass distribution during this study showed a pronounced difference between the WSC in the eastern part of the Fram Strait and the EGC in the western part (Fig. 2 & S3). A temporal transition to the onset of sea ice melt and the development of a spring algal bloom was visible as a freshening of surface waters and increasing chlorophyll concentrations (Fig. 4d). Even though the air temperature started to rise above 0°C from 10 June onwards, a transition in hydrographic and biological parameters was already observed after the storm event on 27 May, with the most pronounced changes on 5 June.

### 4.1 CH$_4$ and N$_2$O Dynamics

Our results show that the surface waters of the Fram Strait acted as a minor overall source of $CH_4$ and $N_2O$ during the spring-summer transition and the onset of sea ice melt in 2023 (Fig. 4a & b, Table 2). Outgassing of $N_2O$ occurred throughout the entire sampling period, while $CH_4$ outgassing was observed at 10 out of 13 stations. Even though the computed sea-air fluxes considered sea ice coverage, some uncertainties remain because SAS enrichment in the SML and the differentiation between ice holes and leads are not included in the parametrization. Complete and partial sea ice cover can significantly influence the gas transfer velocity, primarily due to decreased wind dependency and lower wind fetch near the ice edge, which alters wind-wave interactions (Rutgers Van Der Loeff et al., 2014; Prytherch and Yelland, 2021). For complete sea ice cover, sea-air fluxes are reduced by approximately 90% (Rutgers Van Der Loeff et al., 2014) and for open leads by approximately 30% (Prytherch and Yelland, 2021) compared to widely used parametrizations, and a constant gas transfer velocity ($k_{660}$) of 2.5 cm h$^{-1}$ (Prytherch et al., 2024) was reported.

The parametrization used in our study by Butterworth & Miller (2016) accounts for SIC in the area but does not distinguish between lead and under-ice conditions. Consequently, the results provide flux estimates for the broader sampling area rather than fluxes specific to the open lead. In our study, the difference between $F_{open}$ and $F_{SIC}$ ranged from 17–94%. On average, the sea-air fluxes were reduced by approximately 72% due to sea ice coverage. Although our flux estimates are based on parametrizations and no direct measurements were available, the results are in good agreement with previously measured reductions in sea-air fluxes by SIC in comparable sea ice settings. For high SIC (>80%), our calculations showed an average reduction in gas transfer velocity of 88%, compared to the approximately 90% reduction measured by Rutgers Van Der Loeff et al. (2014). For lower SIC (<80%), the calculated reduction was on average 33%, which is consistent with the approximately 30% reduction measured by Prytherch & Yelland (2021). Since $CH_4$ and $N_2O$ were subject to the same physical drivers, but negative sea-air fluxes (i.e., uptake by the ocean) were only observed for $CH_4$, it is likely that different factors influenced the fluxes.

Another factor influencing the gas transfer velocity is the enrichment of SAS in the SML (e.g., Mustaffa et al., 2020; Pereira et al., 2016). On two sampling days (29 May and 5 June), the SML exhibited SAS concentrations >200 μg Teq L$^{-1}$, which potentially further reduced the gas transfer velocity. Pereira et al. (2016) observed a reduction in gas transfer velocity due to SAS-enrichment by 14–51% in tank experiments, while Mustaffa et al. (2020) found a reduction by 23% for SAS concentrations >200 μg Teq L$^{-1}$ and by 62% in slick conditions (SAS >1000 μg Teq L$^{-1}$) during *in situ* observation. Both days, 29 May and 5 June, represented slick conditions. However, the sampling location of a lead within sea ice represents a unique setting, distinct from open ocean or coastal environments. For example, as the wind speed during sampling on 29 May was high (approx. 10 m s$^{-1}$) and sampling was conducted downwind of the lead, a stronger accumulation of SAS at the sampling location compared to the open lead waters further away from the ice edge, and therefore an overestimation of SAS concentrations, is likely. Since all lead samples were collected downwind, a comparison in consideration of the wind speed can still represent trends in SAS concentrations. The wind speed on 5 June was low (approx. 3 m s$^{-1}$) and therefore a reduction in sea-air flux caused by high concentrations of SAS in the SML likely occurred on this day, but cannot be quantified with the available data.

In general, the observed $CH_4$ and $N_2O$ concentrations were higher than those reported in previous studies from the Fram Strait. The mean $CH_4$ saturations relative to the atmosphere found in this study (110±14%), are slightly higher than those reported by Rees et al. (2022) in the Fram Strait during July 2018, where the central part of the Fram Strait was close to atmospheric equilibrium while oversaturation of up to 132% was present in the surface

waters of the Greenland Shelf (not investigated by the present study). Similarly, $N_2O$ saturations in the present study ($112\pm5\%$) are slightly higher than the values found by Rees et al. (2022) (95–107%, with the highest values found on the Greenland Shelf) and Rees et al. (2021) (89%). As these studies were conducted in July and June, respectively, the difference in $CH_4$ and $N_2O$ saturation could be explained by the reduction of complete sea ice coverage with ongoing sea ice melt, which allows gas exchange between the sea surface and the atmosphere, and by dilution of surface waters with meltwater from sea ice. In a study by Damm et al. (2015b), the ice-free Atlantic Water in the central Fram Strait during July and August 2008 was found to be undersaturated in $CH_4$ or close to equilibrium, while a surface $CH_4$ concentration maximum of 9 nmol $L^{-1}$ was observed in the ice-covered and nitrate-limited Polar Water approximately 190 km west of our study area.

**4.2 Surface dynamics and the onset of sea ice melt**

In the EGC, the first sign of meltwater appeared on 20 May, indicated by a drop of $>0.5$ g $kg^{-1}$ in surface salinity and a $0.3°C$ increase in surface water temperature (Fig. 4c), likely driven by air temperatures rising above $0°C$. Despite this, no changes in solar radiation, chlorophyll, or trace gas concentrations were detected. In contrast, within the waters of the WSC, a temporal transition occurred, starting on 28 May. Here, a continuously decreasing trend in surface salinity and a more variable increasing trend of surface water temperatures were observed. Chlorophyll concentrations increased sharply from 28 May to 1 June and stayed high until the end of the study. Both indicates the onset of sea ice melt and an early algal bloom, likely triggered by a period of high downwelling shortwave radiation (26 May to 6 June) and the storm event (24 to 27 May) bringing nutrient-rich waters to the surface (Fig. 3). Inorganic nutrient concentrations ($PO_4^{3-}$ and $NO_x$, Fig. S4) in surface waters showed a maximum on 27 May and a subsequent decrease during increasing algal growth, reaching concentrations below the detection limit by 1 June for $PO_4^{3-}$ and 5 June for $NO_x$. Therefore, two regime shifts may have occurred: On 29 May, when the algal bloom began, and on 5 June, when shifts in nutrient, trace gas, and SAS concentrations were observed.

On these two days, an accumulation of SAS was observed in the SML, suggesting increased primary productivity (Castillo et al., 2010). The SML concentrations on 29 May were 1435 µg Teq $L^{-1}$, which is within the range of reported values from slick conditions (Wurl et al., 2009). However, the concentration on 5 June (11788 µg Teq $L^{-1}$) exceeded the typically observed values by approximately an order of magnitude. On the same day, the greatest difference between filtered and unfiltered SAS samples was observed, indicating a high proportion of particulate SAS. Several studies conducted in the AO have found an accumulation of particulate organic material in the SML. Particularly, EPS, which can form gel-like biofilms, were found in Arctic SMLs (e.g., Gao et al., 2012). EPS are used as a cryoprotectant and can be produced by, e.g., diatoms within sea ice (Krembs et al., 2002) and dinoflagellates (Guo et al., 2025). The release of EPS has been observed in melt ponds (Galgani et al., 2016), during brine rejection, and algal blooms below ice (Wurl et al., 2011). Underwood et al. (2010) reported high concentrations of EPS in sea ice, particularly in brine, while Gao et al. (2012) observed EPS accumulation in the SML of Arctic leads, with a high fraction of colloidal EPS. These findings suggest that in our study, EPS may have been released during the onset of sea ice melt and brine rejection. The substantial difference in SAS concentration between the filtered and unfiltered samples, particularly on 5 June, indicates that a large fraction of colloidal EPS accumulated in the SML, consistent with the observations of Gao et al. (2012). The extraordinarily high SAS concentration of 11788 µg Teq $L^{-1}$ likely resulted from a combination of EPS release from melting sea ice, *in situ* production during the phytoplankton bloom, and physical accumulation at the lead side. EPS from the

SML in open leads can become aerosolized by bubble-bursting and contribute to the formation of CCN (Leck and Bigg, 2005; Orellana et al., 2021).

The following section discusses the phytoplankton community composition and its potential impact on SAS concentrations. The first maximum in SAS on 29 May coincides with a high relative abundance of dinoflagellates (Fig. 6), possibly producing EPS, which accumulated at the lead side due to the high wind speed of 10 m s$^{-1}$. On 5 June, the community had shifted to be dominated by the diatom *Mediophyceae*. As diatoms are well documented to be EPS producers in polar oceans (e.g., Krembs et al., 2002), this is likely the reason for the high SAS

concentrations on this day. Since the wind speed on this day was low (3 m s$^{-1}$), less enrichment on the downwind side of the lead compared to 29 May is expected. On 8 June, SAS concentrations were significantly lower, despite a high relative abundance of *Mediophyceae*. This likely resulted from lower chlorophyll concentrations, indicating reduced phytoplankton biomass and consequently less SAS release (e.g., Ortega-Retuerta et al., 2009). These observations suggest that the high concentrations of SAS during this study may have been caused by the release

of EPS, mainly from diatoms and, to a smaller extent, from dinoflagellates, either from melting sea ice or from under-ice algal blooms. Algae of the genus *Phaeocystis (Prymnesiophyceae)*, which are known to produce gelatinous foams and EPS (e.g., Li et al., 2023), seemed to have a less pronounced influence on SAS concentrations.

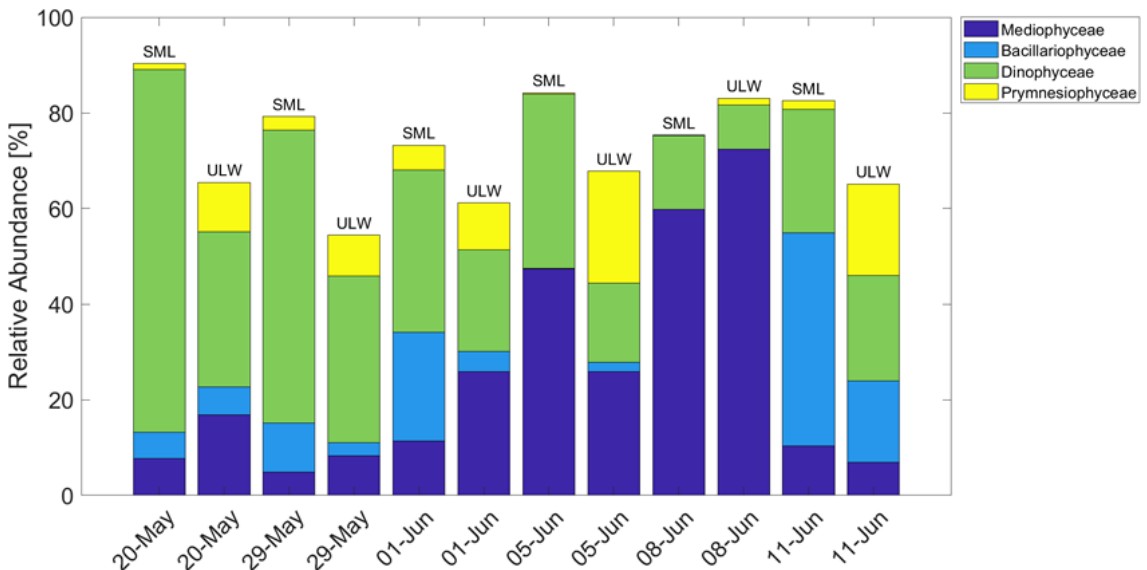

*Figure 6: Relative abundance of selected phytoplankton classes over time in the SML and ULW. Prymnesiophyceae were dominated by Phaeocystis.*

    Given that the onset of sea ice melt and microbial processes can influence both SAS and trace gas dynamics, we

next discuss surface CH$_4$ and N$_2$O concentrations along with possible biotic and abiotic sources and sinks. As we lack data on gene expressions, we discuss potential CH$_4$ and N$_2$O production/consumption pathways based on nutrient availability and bacterial abundances. On 5 June, the shift in SAS concentrations coincided with changes in surface trace gas concentrations, showing an increase in CH$_4$ and a decrease in N$_2$O. As the surface concentrations of both gases show different patterns while exposed to the same physical drivers, different sources

and sinks must have been present. As inorganic nutrients were depleted on 5 June, but organic material from the algal bloom was available, a shift in microbial processes to use alternative nutrient sources may have occurred, although this cannot be confirmed without supporting data such as gene expression or isotopic signatures. For example, pathways such as DMSP degradation (Damm et al., 2008) or methylphosphonate (MPn) cycling (Karl et

al., 2008) have been suggested to contribute to $CH_4$ production. The sharp decrease in $CH_4$ from 1 to 17 m (Fig. S5) suggests a process that is limited to the upper meters of the water column, possibly linked to sea ice melting. Damm et al. (2015a) proposed that brine within sea ice can be strongly enriched with $CH_4$ and be released during sea ice formation and during the onset of basal sea ice melting in spring. Other studies have found that sea ice itself is undersaturated in $CH_4$ and $N_2O$ and that meltwater in general presents a sink of $CH_4$ and $N_2O$ for surface waters, while brine might be oversaturated (Kitidis et al., 2010; Randall et al., 2012; Fenwick et al., 2017). As brine channels can become suboxic and anoxic, they can provide favorable conditions for both anaerobic methane production and denitrification. However, since measurements of trace gas concentrations in both sea ice and brine solution are unfortunately not available from our study, this explanation remains speculative. The highest $N_2O$ concentrations were detected in the surface waters of the EGC, on the Yermak Plateau, and during the ice drift on 8 June. These maxima coincided with the presence of *Nitrosomonas* and *Shewanella* (Fig. S6), which are potential $N_2O$ producers (Cantera and Stein, 2007; Chen and Wang, 2015). The maximum concentrations occurred on the Yermak Plateau on 29 May, which is consistent with previously reported $N_2O$ enrichment in Arctic shelf areas (e.g., Rees et al., 2022). An observed drop in total bacterial abundance on 5 June may suggest a decrease in microbial $CH_4$ oxidation potential, which could contribute to the elevated $CH_4$ concentrations on this day. In the EGC, lower surface saturations, close to equilibrium, were observed and are possibly a result of higher $CH_4$ oxidation rates. However, since data on active gene expression are not available from our study, this is no direct evidence. The drop in both $CH_4$ and $N_2O$ from 9 to 11 June, reaching $CH_4$ undersaturation, was likely caused by dilution with meltwater, as it coincided with rising air temperatures above 0°C, which likely enhanced sea ice melt, particularly surface melt.

**5. Conclusion**

The results of our study in the Fram Strait during spring 2023 revealed that the first hydrographic signs of the onset of sea ice melt and a spring algae bloom coincided in late May and early June before the air temperatures rose above 0°C. A significant shift in biogeochemical parameters was recorded between 29 May and 1 June, which was likely triggered by a combination of a preceding storm event delivering nutrients to the surface and increased solar radiation. This shift included the accumulation of SAS in the SML, particularly in open leads, likely caused by the excretion of EPS by phytoplankton, dominantly diatoms, during the algal bloom. As leads represent an ice-free area, they exhibit both the potential for higher sea-air gas fluxes and greater uncertainties in estimating these fluxes compared to a complete sea ice cover (e.g., Prytherch et al., 2024). We found that surface concentrations of $CH_4$ and $N_2O$ exceeded the atmospheric equilibrium in most cases and therefore the Fram Strait acted as a minor source of $CH_4$ and $N_2O$ during spring 2023. The accumulation of SAS in the SML may have acted as a natural barrier for trace gas emissions due to a reduction in gas transfer velocity. The variability in surface $CH_4$ and $N_2O$ concentrations was driven by both spatial differences and shifts in surface biogeochemistry. The EGC carried more $N_2O$ than the WSC, while $CH_4$ concentrations were slightly lower in the EGC. The variability within the WSC was likely caused by shifts in microbial and sea ice melt processes. Our results emphasize the need for future research on sea-air interactions, considering the presence of the SML in sea ice-influenced areas, to better constrain climate-relevant gas fluxes in rapidly changing polar environments.

Surface trace gas concentration measurements directly from the ice-ocean interface in the AO are rare due to the region's remoteness and challenging weather conditions. Therefore, the presented data are important for accurately quantifying near-surface gas gradients, as traditional CTD rosette sampling only accesses depths of around 1–5 m and is affected by ship-related disturbances. To elucidate the underlying mechanisms of the observed processes, a longer sampling period during the melt season, including sea ice and brine sampling, should be conducted, and stable isotope analysis and metatranscriptomics should be used to gain insights into the microbial processes involved in $CH_4$, $N_2O$, and SAS dynamics.

**Data Availability**

Data and metadata are available at the Bolin Centre for Climate Research (submitted):

CH$_4$, N$_2$O & SAS:

https://doi.org/10.17043/oden-artofmelt-2023-surfactants-1

VMP:

https://doi.org/10.17043/oden-artofmelt-2023-vmp-ctd-1

CTD:

https://doi.org/10.17043/oden-artofmelt-2023-ctd-1

**Author Contribution**

Conceptualization: LAH, JCM. Methodology: LAH, JCM, HWB, DLAM, OW. Investigation: LAH, JCM, JSS. Resources: OW, HWB, JCM, TST. Data Curation: LAH, JCM, JSS. Formal Analysis: LAH. Visualization: LAH. Supervision: DLAM, HWB, OW. Funding Acquisition: OW, THB, HWB, DLAM. Writing – Original Draft: LAH. Writing - Review & Editing: All authors.

**Competing interests**

The authors declare that they have no conflict of interest.

**Acknowledgement**

This work was part of the Early-Career-Scientist program on board the Swedish ice breaker ODEN during the expedition ARTofMELT2023, which was hosted by the Swedish Polar Research Secretariat (SPRS). We would like to thank the crew of ODEN, the co-chief scientists Michael Tjernström and Paul Zieger, and the SPRS-Team for the opportunity of conducting this research and their support. We would like to thank Carolin R. Löscher and Zarah Kofoed (SDU) for conducting trace gas measurements. We acknowledge the use of AI tools (ChatGPT by OpenAI, Claude by Anthropic, and Grammarly) for minor linguistic improvements, such as spelling, grammar, and clarity, but confirm that these tools were not used for content creation or scientific interpretation.

**Financial Support**

The study was funded by the DFG Priority Program 1158 'Antarctic research and comparative investigations in Arctic ice areas' (project EWARP, grant no. 462668354) and the Swedish Polar Research Secretariat (Expedition ARTofMELT2023). TST and JSS would like to gratefully acknowledge funding from The Villum Foundation (23175 and 37435) and The Novo Nordisk Foundation (NNF19OC0056963).

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
