# Peer review of "Biogeochemical Shifts During Arctic Spring: Potential Reduction of CH4 and N2O Emissions Driven by Surfactants in the Sea-Surface Microlayer"

_EGUsphere, 2025_

## Referee Comment (RC1)

**- Review Report -**

**General Comments**

This manuscript presents the first in situ observations of surfactant accumulation in the sea-surface microlayer (SML) of the Fram Strait during the onset of Arctic sea ice melt (spring 2023), coupled with $CH_4$ and $N_2O$ measurements. The study addresses an important knowledge gap: the influence of short-term surface processes on climate-relevant trace gas fluxes in polar oceans. The novelty lies in linking surfactant dynamics, algal bloom development, and greenhouse gas (GHG) emissions, providing new insights into how the SML may act as a natural regulator of $CH_4$ and $N_2O$ fluxes in rapidly changing Arctic environments.

The manuscript is generally well-written, logically structured, and supported by robust datasets from a challenging field campaign. Figures are clear and informative, and the interpretations are scientifically sound. Overall, the study is valuable and has strong potential for publication, but the current manuscript requires several important clarifications and improvements as detailed below.

**Specific Comments**

1. Abstract (p.2, lines 24–31): The conclusion that surfactants reduced $CH_4$ and $N_2O$ emissions is compelling but somewhat overstated. Please add quantitative uncertainty ranges for this reduction with specific numbers.

2. Introduction (p.3, lines 79–87): The introduction nicely describes $CH_4$ and $N_2O$ sources/sinks in the Arctic. However, the information about the role of EPS and algal-derived surfactants is likely to be insufficient here. Please provide more information about the background of EPS in polar environments and their potential to form slicks, and etc.

3. 2.3 section (p.5, lines 135–136): Please clarify why poisoning was done with different $HgCl_2$ volumes for $CH_4$ (125 µL) $vs$ $N_2O$ (50 µL).

4. 2.3 section (p.5, lines 143–147): The time lag between sampling at leads/ice holes and sample poisoning is critical. I consider this the major weakness of the manuscript, and the authors should provide scientifically robust evidence to validate this assumption.

5. 2.5 section (p.7, lines 187–188): The thresholds of >200 and >1000 μg Teq $L^{-1}$ originate from studies conducted in non-Arctic conditions (e.g., Wurl et al., 2011; Mustaffa et al., 2020). Since physical conditions in Arctic leads differ substantially, the authors should provide justification for applying these temperate thresholds to Arctic environments.

6. 4.2 section (p.13, lines 340–350): The authors interpret the June 5 $CH_4$ increase and $N_2O$ decrease as possibly driven by shifts in microbial pathways. However, this explanation remains speculative without direct evidence (e.g., microbial rate measurements or isotopic signatures). The authors should present this more cautiously and highlight the need for the validation.

7. Table 2 (p.9): Negative $CH_4$ fluxes were observed at some stations. These are important and should be discussed in more depth.

8. Fig. 5 (p.10): The extremely high SAS concentration observed on June 5 is remarkable, approximately an order of magnitude higher than typical values. Moreover, the discrepancy between filtered and unfiltered samples is unusually large. This phenomenon requires a more thorough explanation.

9. 4.1 section (p.11, lines 265–275): The authors apply the parametrization of Butterworth & Miller (2016), which was developed for open oceans and marginal ice zones. Please elaborate on the limitations of using this parametrization in semi-ice-covered Arctic leads, where turbulence, wind fetch, and ice-edge effects may differ substantially.

---

## Referee Comment (RC2)

This paper reports on a suite of biogeochemical parameters measured during an ice drift experiment in the Fram Strait. It is an interesting paper, because the conditions under which the measurements were made are challenging, and therefore underrepresented in the literature. The suite of measurements is also rare, with these particular measurements rarely measured together. So... we have a paper with with a rare combination of measured variables, in a rarely measured environment.

It is also very well written. The paper follows a logical flow, and the results and discussion are well supported with good tables and figures. The authors clearly understand the limitations of their study, and do not attempt to overstate the significance of their results.

The only negative comment I have about this paper is that the connections between the different data types are not particularly strong. For example, there is clearly a theoretical link between dissolve greenhouse gas measurements and measurements of surface film concentration. But, with no direct flux measurements (e.g. by a chamber or eddy covariance) there is no analytical connection between GHG concentrations and SAS concentrations, and so we don't actually learn anything new about how these things interact. Similarly, there are interesting measurements of chlorophyll, nutrients, and phytoplankton abundances, all of which may impact GHG concentrations and SAS concentrations... But again, there is no strong analytical link between the two datasets.

So really, there is nothing wrong with this paper. And as I mentioned before, it is to the author's credit that they don't try to over-interpret the data and claim links that are unsubstantiated. But the lack of connections will limit the impact of this paper, and the editors will need to decide if it will have enough impact to be published in a strong journal like The Cryosphere.

I would certainly recommend publication, and only have a few minor comments that the authors might wish to address before publication:

Figure 1 and Table 1: I did not find it very easy to cross-reference between these two paper elements. It was difficult to understand where (spatially) certain measurements had been collected because the only reference was through date... Something like station names probably would have been easier.

Lines 170-175: I would have appreciated more detail on both the sampling and analysis of SAS. Here are some of the things I found confusing about this section:

- -How certain are we that the glass plate method is successful in collecting all of the surface film? It seems that some materials might not adhere to it
- -It wasn't clear why the samples were split into filtered and unfiltered bottles
- -I don't understand what the voltammetry technique does... What is the fundamental principle of measurement that allows us to understand the SAS?
- -I also don't understand the principle of addition of Triton X-100.

Figure 3: It is not clear whether all three variables reported here are averages (if so, over what period)? max, min, etc.

Figure 5: The categories in the caption (e.g. Bulk Unfiltered) did not match the description in the text (ULW).

Lines 265-275: This paragraph needs to be revised. At times, it seems like the authors have forgotten that they didn't actually measure a flux, they estimated a flux. For example, they write "the k660 of 2.5 cm/hr reported by Prytherch et al. (2024) is in good agreement with the average kSIC in this study".... This seems to imply that they measured k, when they didn't, they just calculated it. A similar issue emerges when they write "the average reduction in this study was 33%, which is close to the 30% reported by Prytherch and Yelland (2021). The authors just seem to sort of lose the point here a little bit, and I would strongly recommend revising this paragraph.

---

## Author Comment (AC1)

**Review 1**

**General Comments**

This manuscript presents the first in situ observations of surfactant accumulation in the sea-surface microlayer (SML) of the Fram Strait during the onset of Arctic sea ice melt (spring 2023), coupled with  $CH_4$  and  $N_2O$  measurements. The study addresses an important knowledge gap: the influence of short-term surface processes on climate-relevant trace gas fluxes in polar oceans. The novelty lies in linking surfactant dynamics, algal bloom development, and greenhouse gas (GHG) emissions, providing new insights into how the SML may act as a natural regulator of  $CH_4$  and  $N_2O$  fluxes in rapidly changing Arctic environments.

The manuscript is generally well-written, logically structured, and supported by robust datasets from a challenging field campaign. Figures are clear and informative, and the interpretations are scientifically sound. Overall, the study is valuable and has strong potential for publication, but the current manuscript requires several important clarifications and improvements as detailed below.

We sincerely thank Il-Nam Kim for the thoughtful and constructive comments. We greatly appreciate the recognition of the novelty and significance of our study, as well as the suggestions for clarifications and improvements. We have revised the manuscript based on the reviewer's comments and provided our response to each comment in blue italics.

**Specific Comments**

1. Abstract (p.2, lines 24–31): The conclusion that surfactants reduced  $CH_4$  and  $N_2O$  emissions is compelling but somewhat overstated. Please add quantitative uncertainty ranges for this reduction with specific numbers.

Quantification of the reduction of sea-air gas fluxes by SAS in the SML was not possible because direct flux measurements were lacking, and SAS likely accumulated on the downwind side of the lead. Therefore, the SAS concentrations near the ice edge were most likely higher than in the central part of the lead. An estimate based on the measured SAS concentrations may have overestimated the reduction in sea-air fluxes. Consequently, we restricted our interpretation to reporting the observed SAS accumulation in the SML of two leads, which may have contributed to a reduction in gas fluxes. To clarify that we are only referring to a potential reduction in sea-air fluxes, we added the term "potentially" in line 28. Additionally, we revised the title to "Biogeochemical Shifts During Arctic Spring: Potential Reduction of  $CH_4$  and  $N_2O$  Emissions Driven by Surfactants in the Sea-Surface Microlayer" for clarity.

2. Introduction (p.3, lines 79–87): The introduction nicely describes  $CH_4$  and  $N_2O$  sources/sinks in the Arctic. However, the information about the role of EPS and algal-derived surfactants is likely to be insufficient here. Please provide more information about the background of EPS in polar environments and their potential to form slicks, and etc.

We revised this paragraph and added information about the role of EPS in the Arctic, particularly in the context of SAS accumulation in the SML:

"In polar environments, extracellular polymeric substances (EPS) are an important component of the SAS accumulating in the SML, where they contribute to the formation of biofilms (Gao et al., 2012; Orsi et al., 1995). EPS are produced by phytoplankton, primarily diatoms, and by bacteria as cryoprotectants and are therefore abundant in sea ice and brine (Aslam et al., 2012; Krembs et al., 2002; Underwood et al., 2013). During melting, EPS are released into the surface ocean, providing a source of organic carbon (Riebesell et al., 1991; Riedel et al., 2006). Due to

cross-linking of their polymers, which mainly consist of polysaccharides, EPS can form marine gels and aggregates that influence particle sinking rates and act as potential sources of cloud condensation nuclei (CCN) (Orellana et al., 2011; Riebesell et al., 1991; Verdugo, 2012). Additionally, these aggregates can serve as hotspots of microbial activity (Simon et al., 2002)."

3. 2.3 section (p.5, lines 135–136): Please clarify why poisoning was done with different  $HgCl_2$  volumes for  $CH_4$  (125  $\mu$ L) vs  $N_2O$  (50  $\mu$ L).

The different volumes of  $HgCl_2$  were used due to the different sample volumes of  $CH_4$  and  $N_2O$  (50 mL and 20 mL, respectively). The greater sample volume for  $CH_4$  was chosen to avoid a  $CH_4$  fraction below the detection limit of the gas chromatograph.

4. 2.3 section (p.5, lines 143–147): The time lag between sampling at leads/ice holes and sample poisoning is critical. I consider this the major weakness of the manuscript, and the authors should provide scientifically robust evidence to validate this assumption.

We agree that the reviewer raises an important point, since the time lag between sampling and preservation can indeed influence the gas concentrations. We have added more details about the storage between sampling and preservation (lines 144–149), explaining that the samples were kept cold and in the dark, and that samples from CTD casts (immediately poisoned) from the same day showed no significant difference in gas concentration:

"Samples from the ice and off-ship stations were kept cold and in the dark, and returned to the ship within a maximum of two hours after sampling. Back on board, they were immediately poisoned with 125  $\mu$ L (CH4) and 50  $\mu$ L (N2O) saturated aqueous solution of mercury chloride (HgCl2). Samples collected during CTD casts were poisoned immediately after sampling. The stations at which samples were taken both from CTD and ice stations (1 June and 5 June) show good agreement in gas concentration, despite the difference in the time elapsed between sampling and preservation (see Figure 4)."

Under these temperatures, and considering that all other procedures were conducted in accordance with international standards (see Wilson et al., 2018), the microbial rates could be expected to be low. Since the storage was done in cold conditions, solubility changes are not expected. Hence, we can be confident that the measurements are correct despite the lag.

5. 2.5 section (p.7, lines 187–188): The thresholds of >200 and >1000  $\mu$ g Teq L-1 originate from studies conducted in non-Arctic conditions (e.g., Wurl et al., 2011; Mustaffa et al., 2020). Since physical conditions in Arctic leads differ substantially, the authors should provide justification for applying these temperate thresholds to Arctic environments.

We acknowledge the reviewer's concern that these thresholds were derived from non-Arctic studies. We have revised the paragraph to clarify that these values were used as comparative thresholds, but that deviations may occur due to differences in SAS composition and hydrographic conditions in Arctic leads. As no Arctic-specific data on gas transfer reduction by SAS are currently available, we applied these estimates to show that a suppression of sea-air fluxes due to the accumulation of SAS in the SML has been previously quantified, with increasing reduction for higher SAS concentrations. Such effects are therefore likely to occur in the Arctic Ocean as well. Quantifying the actual reduction in gas transfer velocity would require direct flux measurements, which were not available in our study. Following the revised paragraph:

"In previous studies conducted under temperate conditions, SAS concentrations exceeding 200  $\mu$ g Teq L-1 have been associated with a reduction in gas transfer velocities by approximately 60%, while concentrations above 1000  $\mu$ g Teq L-1 have been classified as slicks (Mustaffa et al.,

2020; Wurl et al., 2011). Maximum suppression of gas exchange rates was in a similar range in laboratory and wind-wave tunnel experiments using artificial monolayers and natural seawater (Brockmann et al., 1982; Broecker et al., 1978; Pereira et al., 2018; Ribas-Ribas et al., 2018; Salter et al., 2011). We used these values as a comparative threshold, as no Arctic-specific measurements are currently available. However, deviations from this relationship are possible due to potential differences in SAS composition and wind-wave interactions in Arctic leads."

**Literature in this comment:**

Brockmann, U. H., Huhnerfuss, H., Kattner, G., Broecker, H. C., & Hentzschel, G. (1982). Artificial surface films in the sea area near Sylt 1. Limnol. Oceanogr., 27(6), 1050-1058.

Broecker, H. C., Petermann, J., & Siems, W. (1978). The influence of wind on CO2-exchange in a wind-wave tunnel, including the effects of monolayers. J. Mar. Res., 36(4), 595-610.

Pereira, R., Ashton, I., Sabbaghzadeh, B., Shutler, J. D., & Upstill-Goddard, R. C. (2018). Reduced air—sea CO2 exchange in the Atlantic Ocean due to biological surfactants. Nature Geosci., 11(7), 492-496

Ribas-Ribas, M., Helleis, F., Rahlff, J., & Wurl, O. (2018a). Air-sea CO2-exchange in a large annular wind-wave tank and the effects of surfactants. Front. Mar. Sci., 5, 457.

Salter, M. E., Upstill-Goddard, R. C., Nightingale, P. D., Archer, S. D., Blomquist, B., Ho, D. T., ... & Yang, M. (2011). Impact of an artificial surfactant release on air-sea gas fluxes during Deep Ocean Gas Exchange Experiment II. J. Geophys. Res. Oceans, 116(C11).

6. 4.2 section (p.13, lines 340–350): The authors interpret the June 5  $CH_4$  increase and  $N_2O$  decrease as possibly driven by shifts in microbial pathways. However, this explanation remains speculative without direct evidence (e.g., microbial rate measurements or isotopic signatures). The authors should present this more cautiously and highlight the need for the validation.

The reviewer raised an important point, and we have revised the sentence for more clarity:

"As inorganic nutrients were depleted on 5 June, but organic material from the algal bloom was available, a shift in microbial processes to use alternative nutrient sources may have occurred, although this cannot be confirmed without supporting data such as gene expression or isotopic signatures."

7. Table 2 (p.9): Negative CH4 fluxes were observed at some stations. These are important and should be discussed in more depth.

We have added a sentence in the result section emphasizing the observed  $CH_4$  undersaturation and revised the last paragraph of the discussion, including possible explanations for the undersaturation:

"On 18 May, 8 June, and 11 June, negative fluxes and therefore CH₄ uptake by the ocean were observed."

"An observed drop in total bacterial abundance on 5 June may suggest a decrease in microbial  $CH_4$  oxidation potential, which could contribute to the elevated  $CH_4$  concentrations on this day. In the EGC, lower surface saturations, close to equilibrium, were observed and are possibly a result of higher  $CH_4$  oxidation rates. However, since data on active gene expression are not available from our study, this is no direct evidence. The drop in both  $CH_4$  and  $N_2O$  from 9 to 11

June, reaching  $CH_4$  undersaturation, was likely caused by dilution with meltwater, as it coincided with rising air temperatures above 0°C, which likely enhanced sea ice melt, particularly surface melt."

8. Fig. 5 (p.10): The extremely high SAS concentration observed on June 5 is remarkable, approximately an order of magnitude higher than typical values. Moreover, the discrepancy between filtered and unfiltered samples is unusually large. This phenomenon requires a more thorough explanation.

We agree that the SAS concentration, particularly in the unfiltered sample from June 5, is extraordinarily high and that the manuscript would benefit from a more detailed explanation. We have added the following paragraph:

"Underwood et al. (2010) reported high concentrations of EPS in sea ice, particularly in brine, while Gao et al. (2012) observed EPS accumulation in the SML of Arctic leads, with a high fraction of colloidal EPS. These findings suggest that in our study, EPS may have been released during the onset of sea ice melt and brine rejection. The substantial difference in SAS concentration between the filtered and unfiltered samples, particularly on 5 June, indicates that a large fraction of colloidal EPS accumulated in the SML, consistent with the observations of Gao et al. (2012). The extraordinarily high SAS concentration of 11788 µg Teq L-1 likely resulted from a combination of EPS release from melting sea ice, in situ production during the phytoplankton bloom, and physical accumulation at the lead side. EPS from the SML in open leads can become aerosolized by bubble-bursting and contribute to the formation of CCN (Leck & Bigg, 2005; Orellana et al., 2021)."

9. 4.1 section (p.11, lines 265–275): The authors apply the parametrization of Butterworth & Miller (2016), which was developed for open oceans and marginal ice zones. Please elaborate on the limitations of using this parametrization in semi-ice-covered Arctic leads, where turbulence, wind fetch, and ice-edge effects may differ substantially.

We have revised the paragraph to clarify the limitations of the parametrization used and to emphasize that the results are representative of the sampling area rather than specific to open leads.

"The parametrization used in our study by Butterworth & Miller (2016) accounts for SIC in the area but does not distinguish between lead and under-ice conditions. Consequently, the results provide flux estimates for the broader sampling area rather than fluxes specific to the open lead. In our study, the difference between  $F_{open}$  and  $F_{SIC}$  ranged from 17–94%. On average, the sea-air fluxes were reduced by approximately 72% due to sea ice coverage. Although our flux estimates are based on parametrizations and no direct measurements were available, the results are in good agreement with previously measured reductions in sea-air fluxes by SIC in comparable sea ice settings. For high SIC (>80%), our calculations showed an average reduction in gas transfer velocity of 88%, compared to the approximately 90% reduction measured by Rutgers Van Der Loeff et al. (2014). For lower SIC (<80%), the calculated reduction was on average 33%, which is consistent with the approximately 30% reduction measured by Prytherch & Yelland (2021). Since  $CH_4$  and  $N_2O$  were subject to the same physical drivers, but negative sea-air fluxes (i.e., uptake by the ocean) were only observed for  $CH_4$ , it is likely that different factors influenced the fluxes."

---

## Author Response (AR1)

We thank the editor and reviewers for the constructive and insightful comments on our manuscript. We have carefully revised the manuscript and implemented all changes as outlined in our detailed responses to the reviewers. The changes are marked in blue in the author's track-changes file.

We believe that these revisions have significantly improved the clarity and robustness of the manuscript and hope that the revised version is now suitable for publication.

---

## Author Response (AR2)

We thank the editor for the careful reading of the manuscript and for providing several minor comments:

Figure 1 Caption. Spell out VMP before first use.
Table 2 Caption: Explain abbreviations used in first table row (e.g. U10, SIC) also in the caption.
Line 284: Please clarify/rephrase this sentence. The last half sentence on the transfer velocity is hanging a bit. Maybe split in two sentences?
Line 301: ... observationS (use plural?)
Line 358: Rephrase "by the diatom Mediophyceae" as Mediophyceae is not a single species but a class of diatoms

We have revised the manuscript accordingly and uploaded the revised version.